# Vancomycin-Loaded, Nanohydroxyapatite-Based Scaffold for Osteomyelitis Treatment: In Vivo Rabbit Toxicological Tests and In Vivo Efficacy Tests in a Sheep Model

**DOI:** 10.3390/bioengineering10020206

**Published:** 2023-02-04

**Authors:** Nuno Alegrete, Susana R. Sousa, Tatiana Padrão, Ângela Carvalho, Raquel Lucas, Raphael F. Canadas, Catarina Lavrador, Nuno Alexandre, Fátima Gärtner, Fernando J. Monteiro, Manuel Gutierres

**Affiliations:** 1i3S—Instituto de Investigação e Inovação em Saúde, Universidade do Porto, R. Alfredo Allen 208, 4200-135 Porto, Portugal; 2FMUP—Faculdade de Medicina, Universidade do Porto, Alameda Prof. Hernâni Monteiro, 4200-319 Porto, Portugal; 3INEB—Instituto de Engenharia Biomédica, R. Alfredo Allen 208, 4200-135 Porto, Portugal; 4Chemical Engineering Department, ISEP—Instituto Superior de Engenharia do Porto, IPP—Instituto Politécnico do Porto, R. Dr. António Bernardino de Almeida 431, 4200-072 Porto, Portugal; 5EPIUnit—Instituto de Saúde Pública, Universidade do Porto, Rua das Taipas 135, 4050-600 Porto, Portugal; 6Tech4MED™, UPTEC, ASPRELA I, Office-Lab 0.16, Business Campus, R. Alfredo Allen, 455/461, 4200-135 Porto, Portugal; 7Med, Mediterranean Institute for Agriculture, Environment and Development, University of Évora, Pólo da Mitra, Apartado 94, 7006-554 Évora, Portugal; 8IPATIMUP—Instituto de Patologia e Imunologia, Universidade do Porto, Rua Júlio Amaral de Carvalho 45, 4200-135 Porto, Portugal; 9FEUP—Faculdade de Engenharia, Universidade do Porto, R. Dr. Roberto Frias, 4200-465 Porto, Portugal; 10CHUSJ—Centro Hospitalar Universitário S. João, Alameda Prof. Hernâni Monteiro, 4200-319 Porto, Portugal

**Keywords:** drug delivery, nanohydroxyapatite, osteomyelitis, vancomycin

## Abstract

The treatment for osteomyelitis consists of surgical debridement, filling of the dead space, soft tissue coverage, and intravenous administration of antimicrobial (AM) agents for long periods. Biomaterials for local delivery of AM agents, while providing controllable antibiotic release rates and simultaneously acting as a bone scaffold, may be a valuable alternative; thus, avoiding systemic AM side effects. V-HEPHAPC is a heparinized nanohydroxyapatite (nHA)/collagen biocomposite loaded with vancomycin that has been previously studied and tested in vitro. It enables a vancomycin-releasing profile with an intense initial burst, followed by a sustained release with concentrations above the Minimum Inhibitory Concentration (MIC) for MRSA. In vitro results have also shown that cellular viability is not compromised, suggesting that V-HEPHAPC granules may be a promising alternative device for the treatment of osteomyelitis. In the present study, V-HEPHAPC (HEPHAPC with vancomycin) granules were used as a vancomycin carrier to treat MRSA osteomyelitis. First, in vivo Good Laboratory Practice (GLP) toxicological tests were performed in a rabbit model, assuring that HEPHAPC and V-HEPHAPC have no relevant side effects. Second, V-HEPHAPC proved to be an efficient drug carrier and bone substitute to control MRSA infection and simultaneously reconstruct the bone cavity in a sheep model.

## 1. Introduction

Bacterial contamination of bone cavities and subsequent osteomyelitis are frequent unfortunate complications of orthopedic trauma and surgery. The infection rate for total knee and total hip arthroplasty is estimated at 0.92% and 0.88%, respectively [1], and these values may grow to 3% for total knee arthroplasty in the future [2]. Implant-associated infection results in increased patient morbidity and mortality and represents a heavy burden for the health systems. The growth of antimicrobial resistance (AMR) and the emergence of new pathogens have further increased the difficulty in treating these infections, namely, the alarming rising of methicillin-resistant *Staphylococcus aureus* (MRSA) osteomyelitis [3]. Nowadays, this affliction represents more than one-third of adult *S. aureus* infections [4], and the associated morbidity and mortality are superior to those deriving from methicillin-sensitive *Staphylococcus aureus* (MSSA).

Traditionally, osteomyelitis treatment consists of surgical debridement, filling of the dead space, soft tissue coverage, and intravenous administration of AM for long periods (4–6 weeks) [5,6]. However, with this treatment, it is difficult to achieve an effective bactericidal concentration of antibiotics at the infection site, which causes a recurrence rate of infection of up to 30% at 12 months after surgery [7].

To achieve high local antibiotic concentrations, local releasing systems have been studied with interesting results. Gentamicin delivery from poly-methylmethacrylate (PMMA) was described in the seventies with good results [8,9]. However, PMMA presents many disadvantages: first, its polymerization process generates heat, making it unsuitable for many antimicrobials [10]; second, it does not degrade, requiring a second surgical procedure for its removal and further defect reconstruction; third, it releases low antibiotic concentrations over longer periods, inducing bacterial resistance. To overcome these disadvantages, biocompatible and biodegradable delivery substances have been used with success, namely, hydroxyapatite (HA), calcium sulfate (CS), chitosan, and several polymers [11].

Ideally, biomaterials for the local delivery of antibiotics should provide controllable antibiotic release rates, have osteoinductive characteristics, and serve as scaffolds to support osteoconduction and osseous integration [12]. HA can repair bone defects and simultaneously release antibiotics in vitro, suggesting that it may be an adequate vehicle for in vivo antimicrobial delivery [13].

Biocomposites, particularly of nHA and collagen, mimic the composition and structure of bone tissue’s extracellular matrix [14,15,16,17,18], promote bone regeneration [19,20,21], and locally release many biomolecules [20,22]. To improve their drug-delivery ability, many biomaterials, namely, nHA/collagen composites, have been linked to heparin. Heparin enables binding for several biomolecules, namely, antibiotics and growth factors, without compromising their structure, releasing capacity, and bioactivity [23,24,25,26,27]. Vancomycin is a tricyclic glycopeptide antibiotic effective against MRSA [28]. Its intravenous administration has, however, considerable side effects, namely, nephrotoxicity, hypotension, and hypersensitivity reactions [29,30]. Therefore, local release of vancomycin may allow adequate concentrations *on site*, avoiding the systemic effects of intravenous administration.

HEPHAPC is a heparinized nHA/collagen biocomposite previously studied and tested in vitro [31]. In this study, we found that it allowed a vancomycin-releasing profile with an intense initial burst, followed by a sustained release for 19 days, with concentrations always above the MIC for MRSA. In vitro results also showed that cellular viability was not compromised in those circumstances, suggesting that HEPHAPC granules loaded with vancomycin (V-HEPHAPC) may be a promising device for osteomyelitis treatment.

In the present study, HEPHAPC granules were used as a vancomycin carrier to treat MRSA osteomyelitis. First, in vivo Good Laboratory Practice (GLP) toxicological tests were performed in a rabbit model to assure that HEPHAPC and V-HEPHAPC had no relevant side effects and that it was safe to use them as a local treatment. The second objective of this study was to evaluate HEPHAPC and V-HEPHAPC’s efficacy as drug carriers and bone substitutes to control MRSA infection and, simultaneously, to reconstruct bone cavities in a sheep model.

## 2. Materials and Methods

### 2.1. Preparation of the Drug-Delivery System

The preparation and in vitro characterization of the heparinized nHA/collagen composite granules—HEPHAPC—has been described elsewhere [31]. Briefly, to obtain porous ceramic granules, scaffolds were produced using the polymer sponge replication method already described [26]. A high-resilience polyurethane sponge was impregnated with a ceramic slurry prepared using nHA powder, distilled water, and a dispersive agent. After being soaked in the slurry, the sponges were dried at 37 °C and then subjected to two different heat treatment cycles: at 600 °C to burn out the polyurethane foam, and then a maximum plateau at 1050 °C. After heat treatment, the scaffolds were crushed and sieved to obtain the nHA granules with sizes ranging between 2.00 mm and 3.35 mm. For collagen incorporation, type I collagen from bovine Achilles tendon (Sigma-Aldrich) was swollen to prepare 0.5% (*w*/*v*) collagen solution. Afterward, the solution was homogenized and diluted to a concentration of 0.05%. This solution was applied to the nHA granules, and these were placed in a vacuum oven at room temperature for 48 h for collagen inclusion. Collagen crosslinking and heparin immobilization were then simultaneously performed. The crosslinking reaction and heparin immobilization were performed for 2 h at 4 °C. Finally, after the washing steps, the heparinized nHA/collagen granules (HEPHAPC) were dried for 24 h at room temperature. To produce V-HEPHAPC granules, 20 mg of HEPHAPC granules were submerged in Eppendorf tubes with 1 mL of vancomycin (HIKMA Farmacêutica, SA) aqueous solution, with a 50 mg/mL concentration. Vancomycin was adsorbed on the HEPHAPC granules for 2 h at 37 °C and 120 rpm. Afterward, the remaining vancomycin solution was removed, and the V-HEPHAPC granules were transferred to new Eppendorf tubes.

### 2.2. In Vivo GLP Toxicological Tests

In vivo GLP subchronic tests (90 days) were carried out using rabbits as the animal model to check for the potential toxicity of HEPHAPC and V-HEPHAPC according to the ARRIVE guidelines [32].

Fifteen male rabbits obtained from Velaz s.r.o. were divided into one control group (Group C) and two treatment groups (Group G1 and G2). Application of HEPHAPC was completed by surgery into both femurs. Three implant beds for each femur were prepared (2 mm × 6 mm).

#### 2.2.1. Implantation Procedure

The animals were fasted overnight for approximately 12 h prior to the implantation/surgery. Medetomidine/midazolam/butorphanol was used for premedication to provide the optimum conditions for anesthesia. Ketamine was used as a general anesthetic. All necessary precautions were taken to ensure the surgical aseptic application of the implants into the femur. The surgical site, the medial aspect of the femur, was prepared by clipping the fur on both hind limbs from the knees to above the hips. The skin was incised, and the subcutaneous and muscle tissue were dissected all the way down to the bone. In the central diaphysis of each femur, three implant beds with a diameter of 2 mm and a depth of 6 mm were created approximately 1 cm apart. Implants (2 mm × 6 mm) were inserted into the prepared defects (Figure 1).

For each rabbit of group G1, 6 implants with 120 mg of HEPHAPC granules without vancomycin were put in place (3/femur) for a total dose of 720 mg/animal. For each rabbit of group G2, 6 implants with 120 mg of V-HEPHAPC granules were put in place (3/femur) for a total dose of 720 mg/animal. Rabbits in Group C received no material; only bone defects were created, similar to the other treatment groups.

The day of surgery was designated Day 1 of the study. The observation period was planned for 90 days after the implantation. After surgery, food and water were provided ad libitum.

#### 2.2.2. Clinical Observations and Mortality

All animals were observed for clinical signs, morbidity, or mortality once a day during acclimatization and twice a day during the study periods.

Clinical observations included signs of toxicity; changes in the skin, fur, eyes, and mucous membranes; secretions and excretions; and autonomic activity (lacrimation, piloerection, pupil size, and unusual respiratory pattern). Changes in gait, posture, and response to handling, as well as the presence of clonic or tonic movements, stereotypes (excessive grooming, repetitive circling), or bizarre behavior (self-mutilation, walking backward), were also monitored.

Observations were carried out according to SOPs SN-TOX-00 and SN-TOX-09.

All animals were observed for any tissue reaction at the implantation sites once a day during the entire study period. All the animals were individually weighed at their arrival, before the implant application, then weekly, and ultimately before their necropsy. Individual body temperature measurement (rectally) was performed before the surgical procedure, then weekly, and finally before their necropsy. Individual food consumption was recorded weekly.

#### 2.2.3. Clinical Pathology—Hematology and Clinical Chemistry

Blood samples for hematology and clinical chemistry were collected from all animals on Week 1 (Examination 1), on Week 2 (Examination 2), on Week 4 (Examination 3), and on Week 13 before necropsy (Examination 4). The animals fasted for approx. 12 to 18 h before blood sampling, but water was provided ad libitum.

Blood samples were taken by venipuncture from *vena saphena* into Vacuette tubes (Greiner bio-one) containing K3EDTA (hematology), sodium citrate (coagulation), and TAPVAL (without anti-coagulant for serum biochemistry).

ABX PENTRA 60 C+ was used to determine hematology parameters from whole blood samples with K3EDTA. Plasma samples from sodium citrate blood samples were obtained by centrifugation at 4000 rpm for 15 min, and coagulation parameters were determined by Coagulometer STart 4.

Smears for differential leucocyte count were performed (stained by May–Grünwald and Giemsa–Romanowski) and microscopically analyzed.

Serum samples were obtained by centrifugation at 6000 rpm for 15 min. Serum for clinical chemistry was transferred into appropriately labeled and sealed Eppendorf tubes and frozen at −20 °C or below until transport to analyses (SOP-HEM, SOP-BCH).

Biochemical parameters of serum samples were obtained by Dimension Vista^®^ 1500 (Siemens Healthcare Diagnostics, Erlangen, Germany). Raw data results were delivered to the MediTox s.r.o. Laboratory of Hematology and Biochemistry. Results evaluation, reporting, and archiving of Test Site raw data were performed by the Test Facility (Laboratory of Hematology and Biochemistry).

Monitored parameters are available as Appendix A.

#### 2.2.4. Urinalysis

The urine from the animals was collected by cystocentesis during the necropsy on day 91, and the total volume was recorded. Macroscopic analysis and semi-quantitative biochemical analysis were performed on Analyzer Urilyzer^®^ 100 (Pro) (Analyticon^®^ Biotechnologies AG, Lichtenfels, Germany), using strips Combi Screen PLUS (Analyticon^®^ Biotechnologies AG, Lichtenfels, Germany).

The parameters controlled were volume, appearance, color, specific gravity (SG), pH, protein (Pro), bilirubin (Bil), urobilinogen (Ubg), erythrocytes (Ery), ketones (Ket), nitrite (Nit), leucocytes (Leu), and glucose (Glu) using the Urilyzer^®^ 100 (Pro) and Combi Screen PLUS (Analyticon^®^ Biotechnologies AG, Lichtenfels, Germany) method.

#### 2.2.5. Terminal Observation

All animals survived throughout the experimental period. On day 91, they were euthanized with pentobarbital and received a complete post-mortem examination. A full set of tissues was collected and fixed, as referred to below.

#### 2.2.6. Necropsy

All the animals were weighed and thoroughly examined. The external surface and all orifices of the body were examined. The cranial, thoracic, and abdominal cavities were opened and macroscopically examined. Implant sites were macroscopically examined for alterations to the structure. For better identification, each implantation site in both femurs was specified as number 1–3 (from distal to proximal direction) and by the letter L or R, respectively, according to the left femur or right femur. Any abnormalities were recorded with details of their location, color, shape and size. Specific organs were weighed after fat and other contiguous tissues were removed (Table 1). Contra-lateral organs were weighed together. Organ weights and terminal body weights were used to calculate organ-to-body weight ratios. The tissues were preserved in 4% neutral buffered formaldehyde. The eyes, optic nerves, testes, and epididymis were fixed in Davidson’s fluid for 24 h and then transferred to 4% neutral-buffered formaldehyde.

#### 2.2.7. Histological Technique

All collected tissues were processed, wax embedded, and cut at a nominal thickness of approx. 5 μm, then stained with hematoxylin and eosin. Bone and implantation sites were decalcified with formic acid (SOP PAT).

#### 2.2.8. Histopathology

Local biological effects after intra-osseous implantation were determined by the semi-quantitative scoring system according to ISO 10993-6. Histological characteristics, such as capsule formation; inflammation; and the presence of polymorphonuclear cells, giant cells, plasma cells, or material degradation were evaluated for each implantation site. Full histopathology was carried out on all preserved organs and tissues.

The mean irritation score for each animal was calculated according to a semi-quantitative evaluation system. The group mean value of the numerical score of irritation was calculated for the treated groups and the control group. The difference between the mean values of the treated and the control groups indicates the Irritation Index of the Test Item. The negative difference was determined as 0. The values of the Irritation Index are stated according to ISO 10993-6. Group G1′s mean value for the Irritation Index was calculated from only four animals.

### 2.3. In Vivo Sheep Efficacy Testing

#### 2.3.1. Osteomyelitis Induction

##### Bacterial Strain and Inoculum Preparation

To create the infection, a strain of MRSA was used. Bacteria were subcultured on a Tryptic Soy Agar (TSA) plate (at −70 °C storage) and incubated overnight at 37 °C. Fresh colonies were transferred to 100 mL Brain Heart Infusion broth and grown at 37 °C until the cells reached the exponential growth phase. The cells were then centrifuged at 2000 rpm for 10 min, and the pellet was washed in phosphate-buffered saline (PBS). The wash step was repeated twice, and the final pellet of cells was resuspended in 5 mL PBS. The viable cells were enumerated by serial dilution. The desired concentration (10^9^ CFU/mL) of bacterial inoculum was achieved by diluting the washed cell concentrate with the appropriate volume of PBS.

##### Experimental Procedure

A total of 12 adult (3 to 6 years old) female Merino sheep, weighing between 41 and 70 Kg, were used in this experimental phase. The animals were allowed to acclimatize for two weeks in the UÉ facilities, housed freely in an outdoor field, with food and water ad libitum and a natural circadian cycle.

Osteomyelitis of the right distal femoral metaphysis was induced following the protocol described by Sinclair [33]. Briefly, the animals fulfilled a fasting time of 24 h before anesthetic induction. For pain control, a transdermal patch of fentanyl was applied 12 h before surgery (2–3 μg/kg). Animals were pre-medicated with midazolam (0.5 mg/kg IM) and ketamine (4 mg/kg IM). The induction of anesthesia was achieved with a mixture of sodium thiopental (5%) at 5 mg/kg IV and propofol at 5 mg/kg IV. All animals had orotracheal and oroesophageal intubation. Maintenance of inhalant anesthesia was accomplished with 1.5% isoflurane.

After the induction of general anesthesia, the medial aspect of the right knee was aseptically prepared: trichotomy was performed, and the skin was scrubbed with chlorhexidine soap (4%) and ethanol solution. Surgical drapes were then placed, allowing access to the medial femoral condyle. A medial incision was made on the anterior border of the medial collateral ligament, and the fibers of the vastus medialis muscle were bounced until the identification of the bony plane. By digital palpation, the adductor tubercle was identified and used as an anatomical reference point. The bone was perforated with an 8 mm drill until 25 mm deep. A cylindrical stainless-steel plug 20 mm long and 8 mm in diameter (Figure 2) was then placed in the bone hole. The plug’s interior was hollow, and the wall contained 16 holes distributed in 4 layers, each with 4 holes at 90° from each other for communication with the outside environment. At one end of the implant, a locking screw was placed, which simultaneously accessed the implant cavity and allowed its attachment to the bone.

Of the 12 animals, 9 were randomly subjected to infection. To induce the infection, the entire implant lumen was filled with 200 microliters of the previously prepared suspension of MRSA. On the remaining three animals, the lumen of the implant was filled with saline solution. The implant cavity was then closed with the screw cap, simultaneously attaching it to the bone. Throughout the surgery, all animals were constantly monitored for vital signs and evidence of pain.

Post-operatively, animals were administrated flunixin meglumine at 1 mg/kg SC for 3–4 days and were evaluated for any evidence of pain. If necessary, administration of IV buprenorphine (0.005–0.01 mg/kg SC every 4–12 h) was performed. The fentanyl patch was removed after 72 h post-surgery.

The implant was maintained on site for 2 weeks, monitoring the development of infection through clinical evaluation of the animals (general well-being and limping) as well as local signs of infection.

Two weeks after the implant placement, all animals were subjected to X-ray imaging and laboratory blood tests and subjected to a second surgical procedure for implant removal and cavity treatment, depending on the experimental group and according to established protocol. The anesthetic protocol was the same as described for the first surgery. The infected animals were randomly assigned to one of three groups: in Group A (*n* = 3), the implant was removed, and the bone cavity was washed with saline; in Group B (*n* = 3), after implant removal and washing, the bone cavity was filled with HEPHAPC; in Group C (*n* = 3), the bone cavity was filled with V-HEPHAPC. Non-infected animals—Group D (*n* = 3)—were subjected to the same surgical procedure, and the bone cavity was filled with HEPHAPC. All implants and tissue samples collected from bone cavities were analyzed for the presence of bacteria. Selective culture media were used to confirm the presence of the original bacteria and to exclude contamination with other agents. After this second procedure, animals were observed for signs of infection (systemic and local) and monitored for radiographic and analytical evolution of infection. All animals were euthanized eight weeks after the second surgical procedure, and distal femur samples were obtained for the subsequent evaluation of infection healing and biomaterial osteointegration.

#### 2.3.2. Clinical and Macroscopic Evaluation

All animals were evaluated every week after the first surgical intervention until euthanasia for weight and behavior changes, limping, and local signs of infection. Quantification of local signs of infection was made according to the Rissing scale in five degrees [34].

#### 2.3.3. Blood Tests

Blood samples were collected at the time of implant removal and every four weeks until euthanasia. The analyzed parameters were the white blood cell count (WBC), hemoglobin, C-reactive protein (CRP), and erythrocyte sedimentation rate (ESR). Blood levels of vancomycin were also evaluated in all Group C animals.

#### 2.3.4. Microbiological Analysis

A tissue sample was collected from every bone cavity after removing the implant and again at euthanasia. It was inoculated onto blood–agar and mannitol plates and incubated at 37 °C for 24 h. Any isolated microorganisms were identified.

Morphological and biochemical characteristics and antibiotic sensitivity were evaluated. Antibiotic inhibition criteria were used to ensure that the isolated microorganism was identical to the original *inoculum*.

#### 2.3.5. X-rays

A radiographic assessment was carried out on all the animals. Radiographs were taken at the time of implant removal and every four weeks until euthanasia. Posterior–anterior and lateral views were acquired. For X-rays, digital films (DLR Cassette, Digiscan 2H/2C, Siemens) and a Mobilett Plus X-ray unit (Siemens AG, Erlangen, Germany) were used. To assess the development of infection, the presence of a radiolucent halo on the periphery of the implant, representing osteolysis, was evaluated and classified as 0 (no radiolucent halo) or 1 (presence of halo). After implant removal, radiological evaluation was based on Norden criteria (sequestrum formation, periosteal reaction, destruction of bone, and extent of involvement) [35].

#### 2.3.6. Micro CT Analysis

The analysis was performed during the bone-mineralized phase, which has lower attenuation to X-rays than the implanted material, but higher attenuation than soft tissue or marrow. Due to implanted material with very high X-ray attenuation, the soft and marrow tissues did not have sufficient resolution to be considered in the present analysis. In this micro-CT characterization, the softer material (or material with lower X-ray attenuation) was the mineralized bone, while the harder material was the implant. The same image-processing algorithm was performed for both the sample and control groups.

The microstructure and architecture of the granular structures, namely, porosity (space between two ceramic particles, as well as two trabecular bone structures), mean pore size, and interconnectivity, were assessed by micro-CT analysis after 3D reconstructions. Dry samples of each formulation were scanned using a high-resolution micro-CT (SkyScan 1272, Bruker Corporation, Billerica, MA, USA). The hydroxyapatite microparticles profile was traced, and mean pore size and interconnectivity were quantified. Then, 3D projections of the specimens were performed. The 3D structures were acquired with a pixel size ranging from 9 to 18 µm. Approximately 400 to 550 projections were acquired over rotation angles of 180° or 360°, with rotation steps of 0.45° or 0.68°, respectively. Data sets were reconstructed using a standardized cone–beam reconstruction software (NRecon 1.6.10.2, Bruker). The output format for each sample was bitmap images. A representative data set of the slices was segmented into binary images with a dynamic threshold of approximately 100–255 for hard ceramic phase analysis and 30–100 for soft polymeric phase analysis (grey scale values were optimized per sample and analysis). Adjustments to these thresholds were performed according to sample formulation and materials’ X-ray sensitivity. Then, the binary images were used for morphometric analysis (CT Analyser, v1.15.4.0, Bruker) and to build the 3D models (CTvox, v 3.0.0, Bruker). When needed, samples were vertically oriented in DataViewer (v1.5.2.3, Bruker) before proceeding to CT Analyser and CTvox.

#### 2.3.7. Histopathological Analysis

Explants were fixed in 10% *v*/*v* neutral-buffered formalin (Bio Optica Milano S.p.A., Milan, Italy) for at least 7 days. The samples were washed in deionized water, and the areas of interest (areas that contained the inserted implant) were cut using an electric autopsy saw and decalcified for at least 4 days in an EDTA-based solution (Milestone™, Shelton, CT, USA). The decalcification status was checked daily. Samples were then histologically processed in an automated system and embedded in paraffin. Then, 4–5 µm sections in silane-coated slides were taken from each bone specimen using a microtome (ThermoScientific™, Waltham, MA, USA) and stained with Hematoxylin and Eosin (ThermoScientific™, Waltham, MA, USA). Tissue sections were dewaxed in xylene and hydrated through decreasing concentrations of alcohol until 100% water for 5 min. Samples were then incubated in hematoxylin (Richard-Allan Scientific™, Waltham, MA, USA) for 5 min, washed and differentiated, incubated in alcoholic eosin (Richard-Allan Scientific™, Waltham, MA, USA) for 2 min, dehydrated, and mounted. Representative slices from the previous osteomyelitis defect and the interface between bone and biomaterials were analyzed, and the presence of inflammation (acute and chronic), bone necrosis, and new bone formation were evaluated.

#### 2.3.8. Statistical Analysis

Groups B (HEPHAPC) and C (V-HEPHAPC) were compared at baseline and at each of the predefined time points regarding the following outcomes: macroscopic evaluation (categorized as a Rissing score of 3 or higher vs. lower than 3), microbiological evidence of infection (categorized as yes vs. no), histologic evidence of infection (categorized as yes vs. no), radiographic assessment (categorized as a Norden score of 2 or higher vs. lower than 2), C-reactive protein (in mg/L), and white blood cell count (thousands per microliter). Categorical outcomes were compared using Fisher’s exact test, and continuous outcomes were compared using Mann–Whitney’s test.

## 3. Results

### 3.1. In Vivo GLP Toxicology Studies

No complications were observed during and after anesthesia. All surgical wounds healed well. Each body weight value was within normal ranges for all the groups during the whole experimental period. The body weights were constant or even increased during the observation period. Food consumption was well-balanced in all animals. Body temperature was recorded for all animals and maintained within the range of 38.4–39.6 °C, which corresponds to the physiological values for rabbits. No changes in the hematology and clinical chemistry parameters that could hypothetically be related to the surgical procedure were observed during the study.

At the end of the experimental period, the highest values of protein, bilirubin, and urobilinogen urine concentration were observed for animals having the combination of HEPHAPC and vancomycin. The presence of protein varied between neg and 0.3 g/L in C and G1 groups. Slightly higher values (0.3–5.0 g/L) were observed in the G2 group. Bilirubin occurrence varied between negative and 70 µmol/L in all samples during the study. Higher occurrence compared to the control group was found in the G2 group (70 µmol/L found in 4 of 5 animals). Urobilinogen determination was normal in all of the samples in Groups C and G1, except for 35 µmol/L found in one animal in Group G1. A higher occurrence of urobilinogen was observed in the G2 group (70 µmol/L in 2 animals and 140 µmol/L in 2 animals). The relationship of these findings with vancomycin administration could not be excluded. No changes that could directly be related to the experiments in other urinalysis parameters were observed during the study. Organ weight analysis did not find any treatment-related changes either. Both HEPHAPC and HEPHAPC-V with a dose of 720 mg/animal did not cause macroscopic nor histopathological changes in the rabbits’ liver and kidneys, indicative of a toxic effect.

All bone defects after intra-osseous implantation of the HEPHAPC were completely healed with minimal to marked periosteal fibrosis, similar to the control group. On the contrary, there was a mild to severe amount of foreign body granulomas in the bone marrow around the grains of implanted material. These granulomas were, in most cases, lined by layers of newly formed bone. Implantation of the HEPHAPC-V caused similar findings without considerable differences.

The Irritation Index in the animals treated with HEPHAPC, evaluating local tissue reaction to the HEPHAPC implanted (according to ISO 100993-10) after subscription of the control group, was 7.08 (category of slight irritation). The Irritation Index in animals treated with V-HEPHAPC, evaluating local tissue reaction to the V-HEPHAPC implanted according to ISO 100993-10 after subscription of the control group, was 5.76 (category of slight irritation).

### 3.2. In Vivo Sheep Efficacy Testing

#### 3.2.1. Clinical and Macroscopic Evaluation

All nine animals subjected to infection induction (Groups A, B, and C) showed progressive limping in the two weeks before implant removal. They all showed macroscopic signs of infection according to the Rissing scale [34] (RS 3.00, av 3.00). None of the animals in Group D had any clinical or macroscopic signs of infection. The author (NA) also reported the presence of implant loosening in all the animals graded as 3, contrasting with some difficulty in removing the implants in the animals that were graded 0 at the time of the second intervention. All animals in Groups A and B showed progressive limping and local signs of infection until the endpoint. The average Rissing score progressed for Group A animals from RS = 3.00 to RS = 3.33, and for all Group B animals from RS = 3.00 to RS = 3.67. All animals in Groups C and D showed progressive improvement in limping, returning to normal gait four weeks after surgery. The Rissing score changed from RS = 3.00 to RS = 0.00 in Group C and remained 0 in all Group D animals. In the final week, all animals in Group B had a Rissing score of three or higher, whereas none of the animals in Group C were in this category (*p* = 0.002) (Table 2).

#### 3.2.2. Blood Tests

The total WBC count was not different between the infected and control group (8671 vs. 8463 × 10^3^/μL) (*p* = 0.574) in opposition to C-Reactive protein values, with an increase in the infected animals (1334 vs. 1106) (*p* < 0.001). The mean C-reactive protein remained generally higher in Group B when compared to Group C, although differences were small: 1.16 vs. 1.05 mg/L (*p* = 0.376).

#### 3.2.3. Microbiological Analysis

At the time of the second surgical intervention, it was possible to isolate MRSA, similar to the inoculum, in all nine animals. No bacteria were identified in the three animals of Group D. At endpoints, bacteria were identified in 1/3 animals in Group A and 3/3 animals in Group B. No bacteria were found in Group C and D animals. Histological evidence of infection was present in all animals in Groups A and B and absent in all animals in Groups C and D.

#### 3.2.4. X-rays

At the time of the second intervention, a radiolucent halo around the metallic implant was present in all nine animals subjected to osteomyelitis induction (9/9). It was absent in all animals in Group D (Figure 3).

Norden classification was used to characterize the radiological changes between the second intervention and endpoint. At endpoint, 2 out of 3 animals in Group A and all animals in Group B had a radiographic score of 2 or higher (average of 4.00 in Group A and 5.17 in Group B). At the endpoint, 2 out of 3 animals in Group C and all in Group D had a radiographic score below 2 (Figure 4) (Table 2).

#### 3.2.5. Micro CT Analysis

A section of the distal femoral metaphysis, including implanted material, was morphologically evaluated by micro-CT. To assess the invasion of host tissue within the implanted material, micro-CT was used. The hard material (colored red) represents the implanted ceramic material, and the soft material (colored green) is the host extracellular matrix found in the defect site. To distinguish both, an X-ray attenuation threshold was applied (soft/green = 30:100 a.a.; hard/red = 100: 255), revealing a larger invasion of host tissue in Groups C and D over 8 weeks. (Figure 5Ai). The identification of the implanted hydroxyapatite is possible not only due to its characteristic shape as porous ceramic granules are typically obtained through polymer sponge replication but also due to its stronger X-ray attenuation than native bone. Group A is a control group with no implanted material. To analyze the integration of the implant in the host tissue, a 2D micro-CT analysis along the Z-axis (defect depth) was performed, which enabled layer-by-layer profiling of the defect zone and quantification of hard (implant) and soft (host tissue) composition. A continuous formation and integration of tissue connected to the implanted material was verified and quantified for Group D, followed by a similar but not continuous profile in B and C (Figure 5Aii). Accordingly, the total free space (total porosity) decreased from Group A to D, as well as the interconnectivity (Figure 5Bi,Bii). The denser material, corresponding to the implanted granules, increased from A to D because A had no implanted material. Overall, the implanted material was still present in the different groups (B to D) at 8 weeks (Figure 5Biii). The trabecular number per millimeter, which is a relevant indicator of trabecular integrity and new trabecular formation, generally increased from A to D, where Group C (the condition with infection, with antibiotics and material) was the most similar condition to control Group D (the condition without infection, with material without antibiotics), as observed in Figure 5Biv. Following the same trend, while the mean trabecular separation (free space in between trabeculae) was decreased in Groups C and D, the mean trabecular thickness was increased in these two conditions compared to A and B (Figure 5Bv,Bvi). In Figure 5C, the distribution of the trabecular thickness is represented per average size, showing that a higher concentration of thin trabeculae was present in Groups A and, in contrast to C and D, where the distribution tended to result in larger trabecular thickness.

#### 3.2.6. Histopathological Analysis

The histological evaluation of the collected specimens for Group A showed the presence of a mononuclear inflammatory infiltrate compatible with chronic phase osteomyelitis, and some *foci* of the polymorphonuclear cell infiltrate, close to or surrounding bone tissue, were evident, also suggesting some signs of acute infection (Figure 6A). In Group B animals (HEPHAPC), there was a predominance of PMN and apparent suppuration, characteristic images of acute infection. Mononuclear cell infiltration associated with the repair process and the formation of granulation tissue was less evident. In addition to these findings, the appearance of macrophages/histiocytes and some foreign body response giant cells was also visible (Figure 6B).

In Group C, V-HEPHAPC was surrounded by a chondroid matrix and, at some points, bone with osteocyte visualization, suggesting the formation of new bone. Looking at the images at higher magnification (100×, 200×, and 400×), we can see very little inflammatory infiltrate, and the ossification can be more clearly seen (Figure 6C). The biomaterial is surrounded by a bone matrix with still-immature bone formation. These images are very similar to those seen in Group D (Figure 6D), where the material was placed without previous infection. In these cases, at the material periphery, the appearance of bone tissue with numerous osteocytes is clearly visible, with zones of calcification and bone maturation.

## 4. Discussion

Chronic osteomyelitis is a challenging problem in common clinical practice. Controlling chronic osteomyelitis, mainly due to MRSA, and dealing with the resulting bone defects are two major problems that orthopedic surgeons have to contend with. In addition to bone infection, there is sometimes contamination of adjacent soft tissues. Whenever this contamination exists, the treatment must be conjoined. For this combined approach, new therapies are emerging [36]. These new approaches will certainly be an indispensable complement to the treatment of osteomyelitis itself. Presently, gold-standard treatment for osteomyelitis is based on a combination of debridement, irrigation with an antiseptic solution, and intravenous antimicrobial administration [37,38]. Local antibiotic concentrations, when administered by the intravenous route, are low, mainly due to anatomopathological changes in bone substance [37]. Bioresorbable materials for local delivery of antimicrobials that might simultaneously solve the problem of dead space and achieve high local bone concentrations of antibiotics are the focus of the current investigation. In previous studies, pure microsized HA has shown slow-releasing antibiotic elution characteristics in vitro, suggesting that it might provide adequate antibiotic delivery in vivo [13], but the concentration of released vancomycin from HA was too low to be detected 10 days after implantation in the rabbit distal femur [39]. In contrast, nHA has a more efficient releasing profile, characterized by an initial rapid delivery (burst) of the anti-microbial followed by a long term sustained release, achieving concentrations above MIC for 42 days [40]. Previous HEPHAPC-V in vitro studies have shown that the vancomycin releasing profile is characterized by an intense initial burst, followed by a sustained release for 19 days with concentrations always above the MIC for MRSA [31], but the in vivo efficacy of this biomaterial to treat chronic osteomyelitis still requires further testing and validation.

The sheep animal model was selected, as sheep have been used for the study of numerous musculoskeletal conditions and diseases, including biomaterial evaluation, as their size, weight, metabolism, and anatomy are the best for preclinical evaluation, thus enabling correlation with human patients [41,42]. Although sheep cortical bone has fewer Haversian canals than human bone, the rate of bone healing approximates that of human values, with a similar pattern of bone ingrowth into porous implants over time [43]. The use of sheep and goat animal models for bone studies is increasing, with reports growing from 5% to 15% in the last 30 years [41,44]. Body weight, bone size, and bone-healing potential are comparable to humans [45,46]. Sheep have already been used as an adequate animal model of contaminated fracture sites to study *S. aureus* infection after internal fixation [47] after open fractures [48,49] and to study contamination after pin tract infections [50,51]. Sinclair et al. [33] developed an ovine animal model of bone contamination to study the prevention of perioperative-device-related infections. Based on this contamination model, the authors developed a stainless-steel implant to promote osteomyelitis similar to endomedular nails and arthroplasty stems (Figure 1).

The time to produce osteomyelitis in this novel animal model was uncertain. Based on previous pilot studies (unpublished data), the authors found that two weeks was enough to develop a bone cavity visible in X-ray imaging and produce implant loosening, as noticed during surgery. Allowing the infection to proceed further would increase the risk of morbidity and mortality among experimental group animals without any additional benefits. The diagnosis of infection was based on a conjunction of factors: animal behavior and macroscopic appearance of the affected knee, X-ray imaging, blood tests, and microbiology analysis.

After the second surgical procedure, we observed that animals in Groups C had a progressive disappearance of limping, in contrast with animals from Groups A and B, suggesting an evolution to cure. At endpoints, animals’ behavior and macroscopic appearance of the knee and bone cuts were very similar between animals of Groups C and D and in contrast with animals from Groups A and B. Analytical and X-ray evolution also showed a similar pattern: animals that had been subjected to implantation of HEPHAPC-V showed progressive evolution to healing, with results similar to Group D animals (not infected). In contrast, animals subjected to debridement and implantation of HEPHAPC (Group B) had progressive analytical deterioration and bone destruction, similar to untreated animals (Group A). The identification of bacteria at endpoints was significantly different between Groups B and C, with 3/3 positive cultures in Group B and 0/3 in Group C, suggesting the efficacy of locally released vancomycin from the nHA composite.

The morphometric characterization of the different animal groups under investigation revealed that the implanted material remained within the targeted bone defect over the 8-week study period in all animal groups (except Group A without material). Treatment Groups C (with infection and V-HEPHAPC) and D (without infection and HEPHAPC) presented a more robust trabecular composition and implant-to-bone integration, supporting the formation of new bone, in contrast with Groups A (with infection and without material) and B (with infection, with HEPHAPC). The trabeculae number and size increased in treatment Groups C and D, and a continuous transition between host hard tissue and implant was detected, meaning that the efficacy of the implanted material to regenerate the host tissue under infection was efficient in the presence of vancomycin.

Current commercially available calcium-sulfate-based products (e.g., OsteoSetTTM (Wright Medical) and Cerament^®^ G and V (Bonesupport)) are antibiotic-eluting implantable beads approved for use in Europe. However, these devices do not use degradable polymers to control antibiotic release, and consequently, they release antimicrobials bound to the surface for just 2–3 weeks. Furthermore, the rapid degradation of calcium-sulfate-based devices can lead to the formation of seromas, pockets of serous fluid with the potential to evolve into abscesses [52]. The degradation of the cements may, on the other hand, lead to a continuous slow release of antibiotics below MIC for very prolonged periods, leading to an increased chance of AMR. Additionally, the lack of interconnective macroporosity on cements does not enable these materials to promote invasion by bone cells and endothelial cells; thus, new bone tissue cannot be formed, as opposed to what was observed in the case of V-HEPHAPC. All data obtained from crossing the information deriving from blood tests, microbiological analyses, X-ray imaging, micro CT, and histology corroborate that after 8 weeks, the MRSA infection was fully under control with the local release of vancomycin (Group C), leading to a total absence of bacteria, as in the case of non-infected samples from the control group (Group D). Bone regeneration was still not completed at that time point despite the observation of new bone formation and new blood vessels in the areas surrounding V-HEPHAPC granules. This indicates the progress of successful bone regeneration.

A parallel study of the potential toxicological effect of V-HEPHAPC was conducted on rabbits under GLP conditions for up to 90 days (subchronic toxicity evaluation). No toxic effects were associated with HEPHAPC or V-HEPHAPC.

A biomechanical evaluation of the strength of the material was not performed in this study. Although there was evidence of new bone formation filling the previously infected cavity, biomechanical studies may be useful to support the clinical applicability of this biomaterial in diaphyseal infections.

Our results show that V-HEPHAPC can successfully treat osteomyelitis in a sheep model, corroborating the in vitro results previously achieved. This therapeutic approach can simultaneously heal bone infections and repair bone defects. The results of a preclinical large animal model (ovine) suggest that HEPHAPC and V-HEPHAPC may be effective alternatives to re-fill bone gaps and treat chronic osteomyelitis in humans, respectively. Future clinical trials are required to prove such capacity. However, the available results suggest that this technology may be more effective than any other presently available.

## 5. Patents

The material used in these experiments has been patented and is registered as P815.5 WO 2021.06.24—Publication WO 2021124108 A1 PCT.

## Figures and Tables

**Figure 1 bioengineering-10-00206-f001:**
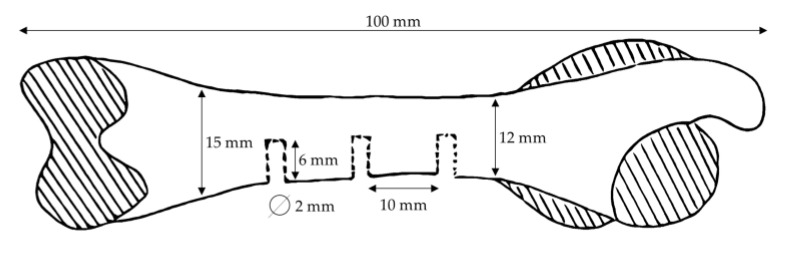
Scheme of implantation site locations on the rabbit femur.

**Figure 2 bioengineering-10-00206-f002:**
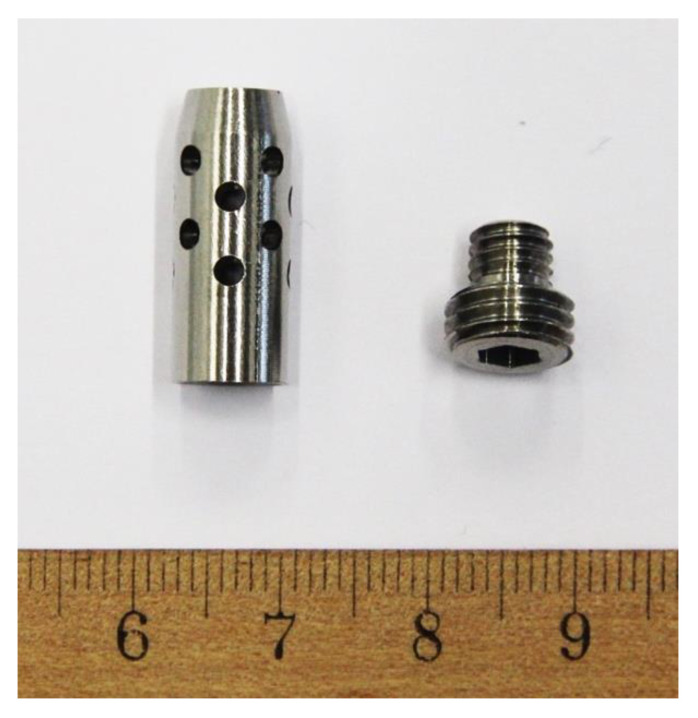
AISI 316 L stainless steel device developed to induce infection in sheep. It consists of a cylindrical hollow stainless-steel plug 20 mm long and 8 mm in diameter. The wall contains 16 holes distributed in four layers, each with four holes at 90° from each other for communication with the outside environment. At one end of the implant a locking screw is placed, which simultaneously accesses the implant cavity and allows its attachment to the bone.

**Figure 3 bioengineering-10-00206-f003:**
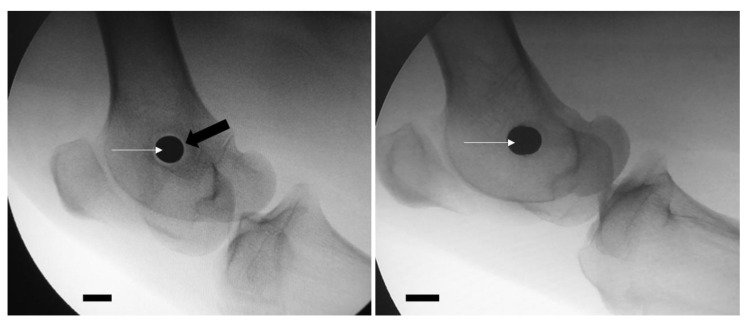
Presence of a radiolucent halo (black arrow) in the periphery of the implant (white arrows) in an animal of osteomyelitis group (**left** image), as opposed to the absence of a halo in the control group (**right** image). Scale bar = 1 cm.

**Figure 4 bioengineering-10-00206-f004:**
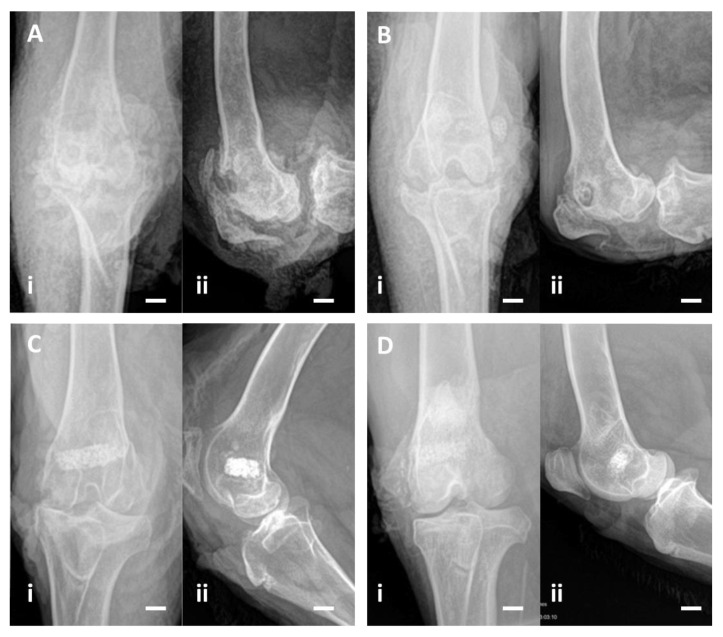
(**A**) In Group A animals, where the osteomyelitis was not treated, an irregular bone cavity is visible, associated with joint destruction. (**B**) In Group B animals, where HEPHAPC was placed (without vancomycin), the biomaterial is surrounded by a radiolucent halo. In some places, the biomaterial is visible outside the cavity initially created, demonstrating its non-integration. (**C**) In Group C animals, implanted with V-HEPHAPC, the biomaterial is integrated, without a radiolucent halo and without extravasation, very similar to the images of Group D. (**D**) In the animals in which the biomaterial was implanted without previous infection (Group D), there is no radiolucency around the implant, showing osseointegration. Scale bar = 1 cm, (**i**) Frontal view; (**ii**) Lateral view of the joint.

**Figure 5 bioengineering-10-00206-f005:**
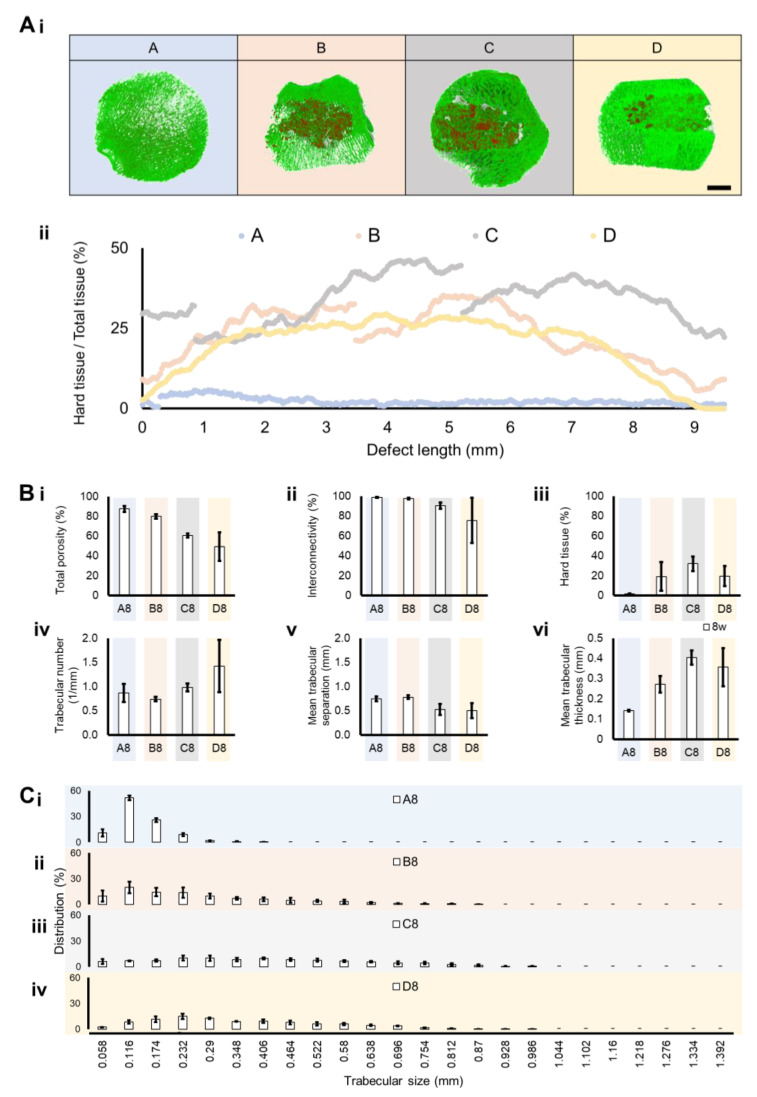
Micro-CT analysis of the implant in the distal femoral metaphysis section and control groups. (**A**) Three-dimensional reconstructions were performed and quantitatively analyzed for each group 8 weeks after surgical intervention. (**i**) Green (threshold of 30:100) and red (threshold of 100:255) represent softer material (equivalent to host tissue) and harder implanted structures, respectively (scale bar = 4 mm). (**ii**) The hard tissue (implant) distribution profile was traced per group, with its percentage represented along a bi-dimensional axis of 9.5 mm. (**B**) A total of 6 morphometric parameters were quantified from the analyzed regions per group after 8 weeks. (**i**) Total porosity, (**ii**) interconnectivity, (**iii**) hard tissue (representing the amount of implant remaining), (**iv**) trabecular number per millimeter, (**v**) mean trabecular separation (meaning the pore size in between trabeculae), and the (**vi**) mean trabecular thickness are represented. (**C**) The trabeculae size distribution was also quantified per average size from 0.058 mm to 1.392 mm, with lower and higher registered trabeculae sizes, respectively, for the following groups (**i**) A—with infection, without material; (**ii**) B—with infection, with material without antibiotics; (**iii**) C—with infection, with antibiotic and material; and (**iv**) D—without infection, with material without antibiotics. Data are shown as mean ± SD (*N* = 2–4).

**Figure 6 bioengineering-10-00206-f006:**
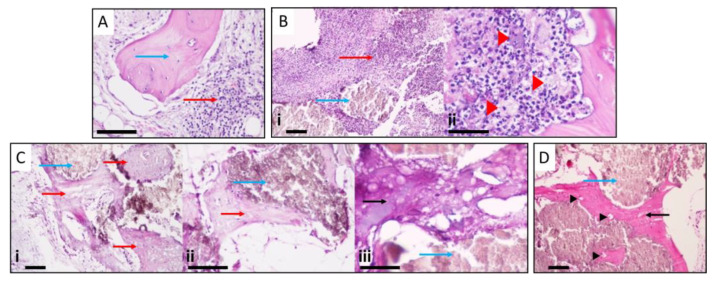
(**A**) Presence of mononuclear inflammatory infiltrate compatible with chronic osteomyelitis and some foci of the polymorphonuclear cell infiltrate (red arrow), close to or surrounding bone tissue (blue arrow), suggesting acute infection. (**B**) The predominance of polymorphonuclear and suppuration (red arrow) around biomaterial (blue arrow). Giant cells are also visible (red arrowheads), suggesting a foreign body reaction. (**C**) V-HEPHAPC (blue arrows) is surrounded by a chondroid matrix (red arrow) and bone, with osteocyte visualization. At higher magnification, very little inflammatory infiltrate and ossification can be clearly seen. In the X200 and X400 images, the biomaterial (blue arrow) is surrounded by chondroid (red arrow) and bone matrix (black arrow) with an area of still-immature bone formation. (**D**) At the periphery of the material (blue arrow), the appearance of bone tissue (black arrow) with numerous osteocytes (black arrowheads) is clearly visible, with zones of calcification and bone maturation. Scale bars = 500 μm (**Ci**); 100 μm (**A**,**Bi**,**Cii**,**Ciii**,**D**)); and 50 μm (**Bii**).

**Table 1 bioengineering-10-00206-t001:** Organs with histopathology examination.

Tissue/Organ	Weight	Fix	Slide	Microscopy
Adrenal glands	×	×	×	×
Aorta		×	×	×
Brain	×	×	×	×
Bone (femur with joint and cartilage)		×	×	×
Cecum		×	×	×
Colon		×	×	×
Duodenum		×	×	×
Epididymides *	×	×	×	×
Eyes (incl. optic nerves) *		×	×	×
Heart	×	×	×	×
Ileum		×	×	×
Implantation sites (3 from each femur)		×	×	×
Jejunum		×	×	×
Kidneys *	×	×	×	×
Lacrimal gland		×	×	×
Larynx		×	×	×
Liver	×	×	×	×
Lungs (incl. mainstem bronchi)		×	×	×
Lymph node (iliac, cervical, mesenteric)		×	×	×
Oesophagus		×	×	×
Pancreas		×	×	×
Payers patches		×	×	×
Pituitary		×	×	×
Prostate	×	×	×	×
Salivary gland (mandibular, parotid) *		×	×	×
Seminal vesicles		×	×	×
Sciatic nerve		×	×	×
Skeletal muscle		×	×	×
Skin		×	×	×
Spinal cord (at three levels)		×	×	×
Spleen	×	×	×	×
Sternum (incl. bone marrow)		×	×	×
Stomach		×	×	×
Testes *	×	×	×	×
Thymus	×	×	×	×
Thyroids (incl. parathyroid) *	×	×	×	×
Tongue		×	×	×
Trachea		×	×	×
Ureters		×	×	×
Urinary bladder		×	×	×
All gross lesions		×	×	×

*—Paired organs; ×—evaluation performed.

**Table 2 bioengineering-10-00206-t002:** Rissing and Norden scores for each animal and group.

Group	Animal	Rissing Score				Norden Score	
		Initial		At Endpoint			
Group A	1	3	3	4	3.33	4	4
	2	3		4		7	
	3	3		2		1	
Group B	4	3	3	4	3.67	6	5.17
	5	3		4		5.5	
	6	3		3		4	
Group C	7	3	3	0	0	0	1
	8	3		0		2	
	9	3		0		1	
Group D	10	0	0	0	0	0	0
	11	0		0		0	
	12	0		0		0	

## Data Availability

Not applicable.

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
