# Peer review of "Vancomycin-Loaded, Nanohydroxyapatite-Based Scaffold for Osteomyelitis Treatment: In Vivo Rabbit Toxicological Tests and In Vivo Efficacy Tests in a Sheep Model"

_bioengineering, 2023, doi:10.3390/bioengineering10020206_

Round 1
Reviewer 1 Report
The manuscript by Alegrete et al entitled « vancomycin loaded nanohydroxyapatite-based scaffold for osteomyelitis treatment : in vivo rabbit toxicological tests and in vivo efficacy tests in sheep model” presents an extensive in vivo study of a biomaterial doped with an antibiotic for the treatment of orthopedic lesions and for the prevention of osteomyelitis.
This is an impressive study, with a lot of analyses of many tissues. It is however somewhat frustrating that despite the huge number of parameters measured in the first section focused on the rabbit model, very few data are provided. Moreover, the data shown to characterize the bone healing process in the sheep model are sometimes not very clear. And the legends and annotations of some figures should be improved to help the reader.
Comments
1. Materials and methods, paragraph 2.1, lines 116-117. What does “in a proportion of 20 mg/ml“ mean?
2. Materials and methods, paragraph 2.2., lines 133-134. It is difficult to figure out how the created bone defect looks like. A scheme or a photograph would help
3. Materials and methods, paragraph 2.2., line 140. The information of 120 mg granules per implant is the most important and relevant and should be placed in first position in the sentence, with the total amount per rabbit in parentheses.
4. Materials and methods, paragraph 2.3.1., lines 235-236. The sentence “when cells reached exponential….phosphate buffered saline” is incorrect.
5. Materials and methods, paragraph 2.3.1., line 238. I do not understand well. The pellet of bacteria was suspended in 5 ml PBS, then plated again, then finally diluted in PBS. How is this possible? How are the cells recovered from the plate? How is the concentrations of bacteria monitored? This part of the protocol is very unclear.
6. Results, paragraph 3.1. I understand that it is not possible, and actually not desirable, to show all the data obtained from the rabbits. Those which do not show any differences between the groups do not need to be shown. However, if some parameters show variations between the groups, they should be shown, for example in a table. This is the case for protein, bilirubin and urobilinogen in urine.
7. Results, paragraph 3.1. lines 385-389. Please show at least one picture illustrating these statements.
8. Results, paragraph 3.2.4.
a. Figure 2. Please show the halo and the implants, using arrows or arrowheads
b. Lines 433-437: a table showing all the scores of the different groups would be useful. In the paragraph results are only partially described.
c. What is the meaning of the sentences lines 438-440? It looks like recommendations for the redaction of the manuscript, or reviewer’s comments.
9. Results, paragraph 3.2.5.
a. I do not understand the part of the sentence line 451-453 “a qualitatively higher incorporation was observed…” The incorporation of what? How is incorporation assessed? What shows the degree of incorporation? This is very unclear. In addition, the sentence suggests that a higher incorporation was observed in groups C, D , A and B, so in all groups!!! This is not consistent. Please clarify.
b. Figure Ai. Please explain what is shown here. This is obviously a bone section, but what section?
c. In the results paragraph as well as in the figure legends it is not clear what the authors mean by “hard tissue” and implant, and what they really measure. There is a confusion between these two terms and it should be clarified. Is the quantification of hard tissue (or implant????) based on the thresholding of microCT images? How is it possible to discriminate between bone tissue and implanted hydroxyapatite with this technique? Also, if only the implanted material is measured in figure Aii, why are the graphs obtained for B, C and D different since these groups all received the same amount of material? Does it mean that there is degradation of the material in some groups?
d. Line 454. What does impregnation mean? With which parameters is “impregnation” assessed?
e. Figure 4. In figure Aii the symbols for the colors used for each group are too small, it is almost impossible to identify the groups. Figure Bi, the legends of axes are too small.
f. All the parameters shown in figure Bi are trabecular bone parameters. This means that thresholding has been done in order to extract bone tissue from the other voxels, for example those representing soft tissue or marrow or those representing the implanted material. It should be explained in details how bone tissue can be identified, extracted and measured from µCT images.
g. What does the “porosity” parameter mean? Is this the porosity of cortical bone?
10. Results paragraph 3.2.6.
In figure 5, arrows or arrowheads or any symbols are needed to show the most important features depicted in the legend (PMN, mononuclear inflammatory infiltrate, bone tissue, giant cells, chondroid matrix, etc….).
11. Discussion.
Line 578. What does “implant bone integration mean?
On what parameters do the authors base their statement of “good” bone integration ?
What does “impregnation of host tissue mean?
Line 581: What does “continuous composition of host hard tissue and implant” mean?
Lines 606-609 : Those three sentences look like recommendations
Author Response
First revision
Vancomycin Loaded Nanohydroxyapatite-based Scaffold for Osteomyelitis Treatment: In Vivo Rabbit Toxicological Tests and In Vivo Efficacy Tests in Sheep Model
Answers to reviewer 1
- Materials and methods, paragraph 2.1, lines 116-117. What does “in a proportion of 20 mg/ml“ mean?
The entire sentence was replaced by: “To produce V-HEPHAPC granules, 20 mg of HEPHAPC granules were submerged in Eppendorf tubes with 1 mL of a vancomycin (HIKMA Farmacêutica, SA) aqueous solution, with a 50 mg/mL concentration.”
- Materials and methods, paragraph 2.2., lines 133-134. It is difficult to figure out how the created bone defect looks like. A scheme or a photograph would help.
An illustration was added to the manuscript.
- Materials and methods, paragraph 2.2., line 140. The information of 120 mg granules per implant is the most important and relevant and should be placed in first position in the sentence, with the total amount per rabbit in parentheses.
The entire paragraph was replaced by “For each rabbit of group G1, six implants with 120 mg of HEPHAPC granules without vancomycin were put in place (3/femur), making up a total dose of 720 mg/animal. For each rabbit of group G2, six implants with 120 mg of V-HEPHAPC granules were put in place (3/femur), making up a total dose of 720 mg/animal. Rabbits in group C received no material; only bone defects were created similar to the other treatment groups.”
- Materials and methods, paragraph 2.3.1., lines 235-236. The sentence “when cells reached exponential….phosphate buffered saline” is incorrect.
The entire paragraph was reviewed. Please, see the next question
- Materials and methods, paragraph 2.3.1., line 238. I do not understand well. The pellet of bacteria was suspended in 5 ml PBS, then plated again, then finally diluted in PBS. How is this possible? How are the cells recovered from the plate? How is the concentrations of bacteria monitored? This part of the protocol is very unclear.
The whole paragraph on inoculum preparation was rewritten and replaced with: “To create the infection, a strain of MRSA was used. Bacteria were subcultured on Tryptic Soy Agar (TSA) plate (from -70ºC storage) and incubated overnight at 37°C. Fresh colonies were transferred to 100 mL Brain Heart Infusion broth and grown at 37°C until the cells reached the exponential growth phase. The cells were then centrifuged at 2000 rpm for 10 min and the pellet was washed in phosphate-buffered saline (PBS). The wash step was repeated twice and the final pellet of cells was resuspended in 5 mL PBS. The viable cells were enumerated by serial dilution. The desired concentration (109 CFU/mL) of bacterial inoculum was achieved by diluting the washed cell concentrate with the appropriate volume of PBS.”
- Results, paragraph 3.1. I understand that it is not possible, and actually not desirable, to show all the data obtained from the rabbits. Those which do not show any differences between the groups do not need to be shown. However, if some parameters show variations between the groups, they should be shown, for example in a table. This is the case for protein, bilirubin and urobilinogen in urine.
Positive results were detailed and the paragraph was rephrased to: “At the end of the study period, the highest values of protein, bilirubin and urobilinogen urine concentration were observed for animals having the combination of HEPHAPC and vancomycin. The presence of protein varied between neg to 0.3 g/l in C and G1 group. Slightly higher values (0.3-5.0 g/l) were observed in G2 group. Bilirubin occurrence varied between negative to 70 µmol/l in all samples during the study. Higher occurrence as compared to control group was found in G2 group (70 µmol/l found in 4 animals of 5). Urobilinogen determination was normal in all of the samples in C and G1 group except for 35 µmol/l found in one animal of Group G1. Higher occurrence of urobilinogen was observed in G2 group (70 µmol/l in 2 animals and 140 µmol/l in 2 animals). The relationship of these findings with the Vancomycin administration could not be excluded. No changes that could directly be related to the experiments in other urinalysis parameters were observed during the study.”
- Results, paragraph 3.1. lines 385-389. Please show at least one picture illustrating these statements.
Once the main goal of this experiment was the evaluation of implanted material safety, histology photos of bone were not recorded. It is possible to go back to the histology lab and take pictures but, due to access limitations in the lab, it will take at least 6 weeks to get these images. Once we do not consider that these images will change the overall result and message of this paper, we kindly ask the reviewer to abdicate this request.
- Results, paragraph 3.2.4.
a. Figure 2. Please show the halo and the implants, using arrows or arrowheads
Figure 2 was updated according to suggestions and renamed “Figure 3”.
b. Lines 433-437: a table showing all the scores of the different groups would be useful. In the paragraph results are only partially described.
A table with Rissing and Norden scores was added to the manuscript
c. What is the meaning of the sentences lines 438-440? It looks like recommendations for the redaction of the manuscript, or reviewer’s comments.
This sentence was removed
- Results, paragraph 3.2.5.
a. I do not understand the part of the sentence line 451-453 “a qualitatively higher incorporation was observed…” The incorporation of what? How is incorporation assessed? What shows the degree of incorporation? This is very unclear. In addition, the sentence suggests that a higher incorporation was observed in groups C, D , A and B, so in all groups!!! This is not consistent. Please clarify.
This information was added to section3.2.5.: “To assess the invasion of host tissue within the implanted material, micro-CT was used. The hard (coulored red) represents the implanted ceramic material, and the soft (coulored green) the host extracellular matrix found in the defect site. To distinguish both, an X-ray attenuation threshold was applied (soft/green = 30:100 a.a.; hard/red = 100:255), revealing a larger invasion of host tissue in the conditions C and D over 8 weeks.”
b. Figure Ai. Please explain what is shown here. This is obviously a bone section, but what section?
The next information was added to the results section and respective figure caption:
“A section of the distal femoral metaphysis including implanted material was morphologically evaluated by micro-CT.”
“Figure 5. Micro-CT analysis of the implant in the distal femoral metaphysis section and control groups”.
c. In the results paragraph as well as in the figure legends it is not clear what the authors mean by “hard tissue” and implant, and what they really measure. There is a confusion between these two terms and it should be clarified. Is the quantification of hard tissue (or implant????) based on the thresholding of microCT images? How is it possible to discriminate between bone tissue and implanted hydroxyapatite with this technique? Also, if only the implanted material is measured in figure Aii, why are the graphs obtained for B, C and D different since these groups all received the same amount of material? Does it mean that there is degradation of the material in some groups?
The quantification of hard material was based on the thresholding of microCT images.
The identification of the implanted hydroxyapatite is possible not only due to its characteristic shape as porous ceramic granules typically obtained through polymer sponge replication, but also due to its stronger X-ray attenuation than native bone.
d. Line 454. What does impregnation mean? With which parameters is “impregnation” assessed?
We changed the term impregnation by integration. The next information was added to the results section to elucidate the analyzed data and respective graphs:
“To analyze the integration of the implant in the host tissue, a 2D micro-CT analysis along the Z-axis (defect depth) was performed, which allows for layer-by-layer profiling of the defect zone and quantification of hard (implant) and soft (host tissue) composition.”
e. Figure 4. In figure Aii the symbols for the colors used for each group are too small, it is almost impossible to identify the groups. Figure Bi, the legends of axes are too small.
Image size was changed
f. All the parameters shown in figure Bi are trabecular bone parameters. This means that thresholding has been done in order to extract bone tissue from the other voxels, for example those representing soft tissue or marrow or those representing the implanted material. It should be explained in details how bone tissue can be identified, extracted and measured from µCT images.
The following explanation was added to the Materials and Methods section, 2.3.6 Micro CT analysis:
“The analysis was performed to the bone mineralized phase, which has lower attenuation to X-rays than the implanted material, but higher attenuation than soft tissue or marrow. Due to the presence of an implanted material of very high x-Rays attenuation, the soft and marrow tissues do not have sufficient resolution to be considered in the present analysis. In this micro-CT characterization, the softer material (or material with lower x-Ray attenuation) is the mineralized bone, while the harder material is the implant. The same image processing algorithm was performed for both, the sample and control groups”.
g. What does the “porosity” parameter mean? Is this the porosity of cortical bone?
The following concept definition was added to the section 2.3.6 Micro CT analysis to facilitate the reader comprehension:
“Porosity means the space between two ceramic particles, as well as two bone trabecular structures. Generally, in the present analysis porosity if the empty space observed under the present x-Ray imaging parameters.”
- Results paragraph 3.2.6.
In figure 5, arrows or arrowheads or any symbols are needed to show the most important features depicted in the legend (PMN, mononuclear inflammatory infiltrate, bone tissue, giant cells, chondroid matrix, etc….).
Figure 5 was updated according to suggestions.
- Discussion.
Line 578. What does “implant bone integration mean?
The implant is integrated into bone when there is full contact between implanted material and adjacent bone tissue, without gap and no fibrous capsule involving it.
The expression was changed to implant-to -bone integration,
On what parameters do the authors base their statement of “good” bone integration?
In Fig. 3 legend, the above-mentioned sentence was replaced by “In the animals in which the biomaterial was implanted without previous infection (Group D), there is no radiolucency around the implant, showing osseointegration.”
What does “impregnation of host tissue mean?
The entire sentence was changed and the above-expression was removed. The sentence was replaced by “Treatment groups C (with infection and V-HEPHAPC) and D (without infection and HEPHAPC), presented a more robust trabecular composition and implant-to-bone integration, supporting the formation of new bone, in contrast with groups A (with infection without material) and B (with infection, with HEPHAPC). The trabeculae number and size increased in treatment groups C and D, and a continuous transition between host hard tissue and implant was detected, meaning that the efficacy of the implanted material to regenerate the host tissue under infection was efficient in the presence of vancomycin.”
Line 581: What does “continuous composition of host hard tissue and implant” mean?
The sentence was replaced by “The trabeculae number and size increased in treatment groups C and D, while a continuous transition between host hard tissue and implant was detected”
Lines 606-609 : Those three sentences look like recommendations
Sentences were removed.
Reviewer 2 Report
Nuno Alegrete et al. provided with their manuscript entitled „ Vancomycin Loaded Nanohydroxyapatite-based Scaffold for Osteomyelitis Treatment: In Vivo Rabbit Toxicological Tests and In Vivo Efficacy Tests in Sheep Model.” a very interesting study.
V-HEPHAPC (HEPHAPC with vancomycin) granules was used as a vancomycin carrier to treat MRSA-osteomyelitis. First, in vivo Good Laboratory Practice (GLP) toxicological tests were performed in a rabbit model, assuring that HEPHAPC and V-HEPHAPC had no relevant side effects. Second, V-HEPHAPC was proved to be an efficient drug carrier and bone substitute to control MRSA infection and, simultaneously, reconstruct bone cavity in a sheep model.
Comments:
Basically, the authors provided a detailed study with some new aspects.
Methods:
In their study, the authors show in detail in 2 animal models the benefit that treatment with HEPHAPC with vancomycin can have. The sizes of the animal groups are well chosen for these experiments.
A biomechanical evaluation of the strength of the specimens is not provided, but this can be dispensed with in the context of the present absence of infection and new bone formation shown histopathologically, although in further experiments this would be an essential factor in being able to treat diaphyseal infections as well.
For the discussion, it would still be interesting to mention newer methods of infection control, such as antibiotic and fibrin spraying, and compare them in context ( e.g. Janko, M., F. Dust, P. V. Wagner, R. Gurke, J. Frank, D. Henrich, I. Marzi, and R. D. Verboket. 2022. 'Local Fixation of Colistin With Fibrin Spray: An in vivo Animal Study for the Therapy of Skin and Soft Tissue Infections', Front Surg, 9: 749600.)
Please revise the whole paper regarding spelling and punctuation, there are still several paragraphs missing (e.g. Author Contributions, delete Patents etc.).
Conclusion:
Very interesting paper that can certainly be accepted after minor revision and modification of the points raised.
Author Response
First revision
Vancomycin Loaded Nanohydroxyapatite-based Scaffold for Osteomyelitis Treatment: In Vivo Rabbit Toxicological Tests and In Vivo Efficacy Tests in Sheep Model
Answers to reviewer 2
A biomechanical evaluation of the strength of the specimens is not provided, but this can be dispensed with in the context of the present absence of infection and new bone formation shown histopathologically, although in further experiments this would be an essential factor in being able to treat diaphyseal infections as well.
The authors thank the recommendation and included the suggestion in the paper discussion.
For the discussion, it would still be interesting to mention newer methods of infection control, such as antibiotic and fibrin spraying, and compare them in context ( e.g. Janko, M., F. Dust, P. V. Wagner, R. Gurke, J. Frank, D. Henrich, I. Marzi, and R. D. Verboket. 2022. 'Local Fixation of Colistin With Fibrin Spray: An in vivo Animal Study for the Therapy of Skin and Soft Tissue Infections', Front Surg, 9: 749600.)
The authors thank the recommendation and included the suggestion in the paper discussion
Please revise the whole paper regarding spelling and punctuation, there are still several paragraphs missing (e.g. Author Contributions, delete Patents etc.).
Language revision was made. Paragraphs on Patents and Author contributions was corrected.